

# Periodically and quasi-periodically driven dynamics of Bose-Einstein condensates

**Pengfei Zhang[1] and Yingfei Gu[2]**

**1** Institute for Quantum Information and Matter and Walter Burke Institute
for Theoretical Physics, California Institute of Technology, Pasadena, CA 91125, USA
**2** Department of Physics, Harvard University, Cambridge MA 02138, USA

## Abstract

We study the quantum dynamics of Bose-Einstein condensates when the scattering length is modulated periodically or quasi-periodically in time within the Bogoliubov framework. For the periodically driven case, we consider two protocols where the modulation is a square-wave or a sine-wave. In both protocols for each fixed momentum, there are heating and non-heating phases, and a phase boundary between them. The two phases are distinguished by whether the number of excited particles grows exponentially or not. For the quasi-periodically driven case, we again consider two protocols: the square-wave quasi-periodicity, where the excitations are generated for almost all parameters as an analog of the Fibonacci-type quasi-crystal; and the sine-wave quasi-periodicity, where there is a finite measure parameter regime for the non-heating phase. We also plot the analogs of the Hofstadter butterfly for both protocols.



# 1 Introduction

The rapid development of experimental techniques in cold atom systems draws a lot of attention to interesting quantum dynamics. An important scenario is to periodically drive the system. When the lattice is modulated periodically in time, one could generate topologically non-trivial band insulators [1–8] or introduce gauge fields with non-trivial dynamics [9–12]. However, quantum dynamics are usually hard to solve. Special cases where large simplification occurs due to symmetry reason is especially valuable.

The driven dynamics of Bose-Einstein condensates (BECs), which has been realized in [13–16], is one of such examples. As pointed out in [17] and later studied in [18,19], the dynamics of the Bose-Einstein condensates under the Bogoliubov Hamiltonian exhibit an SU(1, 1) structure, which, more generally speaking, is a consequence of the bosonic Bogoliubov transformations. The dynamical evolution of the system can then be described by the multiplication of a sequence of SU(1, 1) matrices,[1] acting on two components pseudospin $(a, a^\dagger)$ for bosonic creation and annihilation operators. The observable and the phase diagram can then be obtained by analyzing the corresponding SU(1, 1) matrices. The techniques applied in this work is a close analogy of those in the driven conformal field theory (CFT) in $1 + 1$-D [27–32]. In the CFT case, the system has the Virasoro symmetry and for specific driving Hamiltonians, one could focus on the SL$(2, \mathbb{R}) \cong$ SU(1, 1) subgroup. Another analogy is the dynamics of the scale-invariant gases in the harmonic trap [33,34], as would be discussed in more detail later.

In this work, we consider two types of protocols for both the periodic and the quasi-periodic drivings. The two types correspond to modulating the scattering length[2], which is equivalent to the coupling strength in our BEC setting, by a square-wave or a sine-wave in time[3]. For the square wave protocol, in each period the evolution is determined by multiplying a finite number of matrices while for the sine-wave protocol we, in general, need to solve a Mathieu's equation, however simplifications can be made by Magnus expansion in the near-resonant limit [17]. In both protocols, the long-time dynamical phase diagram is then determined by the matrix element of the SU(1, 1) evolution matrix/Bogoliubov transformation in a single period as in [30]. We then turn to the driving dynamics with quasi-periodicity. For the square-wave quasi-periodic driving, the excitations grow exponentially for almost all parameters, while special points with power-law heating also exist [30] and will be discussed. For the sine-wave type quasi-periodicity, we consider the scattering length to be a superposition of two sinusoidal functions with incommensurate frequencies, where the evolution equation is a quasi-periodic Mathieu's equation [36,37].[4]

The paper is organized as follows: In section 2, we explain the SU(1, 1) group structure of the Bogoliubov Hamiltonian for a driven Bose-Einstein condensate focusing on the Heisenberg evolution of creation/annihilation operators. In section 3, we study the two protocols when the scattering length $a_s(t)$ is a periodic function in time and the quasi-periodic case is discussed in section 4. Finally, we summarize our work and comment on the Bogoliubov approximation in section 5.

---

[1]Similar strategy for the case with SU(2) spin dynamics has been discussed in Ref. [20–26].

[2]One can also consider a time-dependent Zeeman field as in e.g. Ref. [35].

[3]The modulation of the scattering length has also been used for realizing the correlated tunneling in [9].

[4]See e.g. [38] for a review from the perspective of applied mathematics.

## 2  Dynamics of Bose-Einstein Condensates

We consider dilute single-component bosonic atom gases with the s-wave interactions. Taking the time-dependence of the scattering length into account, the Hamiltonian reads:

$$H(t) = \int d\mathbf{r} \left[ \psi^\dagger(\mathbf{r}) \left( -\frac{\nabla^2}{2} \right) \psi(\mathbf{r}) + \frac{g(t)}{2} \psi^\dagger(\mathbf{r}) \psi^\dagger(\mathbf{r}) \psi(\mathbf{r}) \psi(\mathbf{r}) \right]. \tag{1}$$

Here we have set the mass of atoms $m = 1$ for simplicity. At the bare level, the scattering parameter $g(t)$ can be related to the scattering length $a_s$ as

$$g(t) = 4\pi a_s(t). \tag{2}$$

When study dynamics around a specific state with large number of particles in the condensate, we expand $\psi(\mathbf{r}) = \sqrt{n_0} + \delta\psi(\mathbf{r})$, where $n_0$ is the density of condensate particles. To the quadratic order, this gives the Bogoliubov Hamiltonian:

$$H(t) = \int d\mathbf{r} \left[ \delta\psi^\dagger(\mathbf{r}) \left( -\frac{\nabla^2}{2} + g(t)n_0 \right) \delta\psi(\mathbf{r}) + \frac{g(t)n_0}{2} \left( \delta\psi^\dagger(\mathbf{r})\delta\psi^\dagger(\mathbf{r}) + \text{h.c.} \right) \right] + \text{const.}, \tag{3}$$

where the constant term is only a function of the total particle number $N = \int d\mathbf{r}(n_0 + \delta\psi^\dagger\delta\psi)$ and therefore conserved. The validity of using a simple Bogoliubov Hamiltonian to study the dynamics of condensates has been verified by comparing to a careful numerical study which includes the evolution of condensate wave function [17,39,40]. The Hamiltonian can also be written in the momentum space via Fourier expansion $\delta\psi(\mathbf{r}) = \frac{1}{\sqrt{V}} \sum_{\mathbf{k} \neq \mathbf{0}} e^{i\mathbf{k}\cdot\mathbf{r}} a_\mathbf{k}$ with $V$ the system size. Thus, we have

$$H(t) = \frac{1}{2} \sum_{\mathbf{k} \neq \mathbf{0}} \begin{pmatrix} a_\mathbf{k}^\dagger & a_{-\mathbf{k}} \end{pmatrix} \begin{pmatrix} \frac{k^2}{2} + g(t)n_0 & g(t)n_0 \\ g(t)n_0 & \frac{k^2}{2} + g(t)n_0 \end{pmatrix} \begin{pmatrix} a_\mathbf{k} \\ a_{-\mathbf{k}}^\dagger \end{pmatrix} + \text{const.} \tag{4}$$

There are two ways to recognize the $SU(1,1)$ structure in this system (for each $\mathbf{k}$):

1. Using the notation $A_\mathbf{k}^z = \frac{1}{2}(a_\mathbf{k}^\dagger a_\mathbf{k} + a_{-\mathbf{k}} a_{-\mathbf{k}}^\dagger)$ and $A_\mathbf{k}^x = \frac{1}{2}(a_\mathbf{k}^\dagger a_{-\mathbf{k}}^\dagger + a_{-\mathbf{k}} a_\mathbf{k})$, we can rewrite the time-dependent Hamiltonian as a linear combination

$$H(t) = \sum_{\mathbf{k} \neq \mathbf{0}} \left[ \left( \frac{k^2}{2} + g(t)n_0 \right) A_\mathbf{k}^z + g(t)n_0 A_\mathbf{k}^x \right] + \text{const.} \tag{5}$$

Further introducing $A_\mathbf{k}^y = \frac{1}{2i}(a_\mathbf{k}^\dagger a_{-\mathbf{k}}^\dagger - a_{-\mathbf{k}} a_\mathbf{k})$, we have the closed commutation relation for the $\mathfrak{su}(1,1)$ algebra [17,18]

$$[A_\mathbf{k}^x, A_\mathbf{k}^y] = -iA_\mathbf{k}^z, \qquad [A_\mathbf{k}^y, A_\mathbf{k}^z] = iA_\mathbf{k}^x, \qquad [A_\mathbf{k}^z, A_\mathbf{k}^x] = iA_\mathbf{k}^y. \tag{6}$$

2. We can also directly study the evolution of excitation operators $a_\mathbf{k}$ and $a_\mathbf{k}^\dagger$. For a general $g(t)$ and the corresponding unitary $U(t) = \mathcal{T}\exp(-i \int_0^t H(t')dt')$, the time-evolution generates a Bogoliubov transformation between $a_\mathbf{k}^\dagger$ and $a_{-\mathbf{k}}$ as follows

$$\begin{pmatrix} a_\mathbf{k}(t) & a_{-\mathbf{k}}^\dagger(t) \end{pmatrix} = U^\dagger(t) \begin{pmatrix} a_\mathbf{k} & a_{-\mathbf{k}}^\dagger \end{pmatrix} U(t) = \begin{pmatrix} a_\mathbf{k} & a_{-\mathbf{k}}^\dagger \end{pmatrix} \underbrace{\begin{pmatrix} \alpha_\mathbf{k}(t) & \beta_\mathbf{k}(t) \\ \beta_\mathbf{k}^*(t) & \alpha_\mathbf{k}^*(t) \end{pmatrix}}_{=: \mathcal{U}_\mathbf{k}}. \tag{7}$$

In the first step, the unitary $U(t)$ and $U^\dagger(t)$ act on the operators $a_\mathbf{k}$ and $a_{-\mathbf{k}}^\dagger$, while in the last step, the $2 \times 2$ transform matrix $\mathcal{U}_\mathbf{k}$ acts on the basis vector $(a_\mathbf{k}\ a_{-\mathbf{k}}^\dagger)$ (where we

have used the inversion symmetry $\alpha_{\mathbf{k}} = \alpha_{-\mathbf{k}}$, $\beta_{\mathbf{k}} = \beta_{-\mathbf{k}}$ to rewrite the second column of $\mathcal{U}_{\mathbf{k}}$). Further note that the (bosonic) commutation relation $[a_{\mathbf{k}}, a_{\mathbf{k}}^\dagger] = 1$ demands $|\alpha_{\mathbf{k}}(t)|^2 - |\beta_{\mathbf{k}}(t)|^2 = 1$, therefore the Bogoliubov transformation matrix $\mathcal{U}_{\mathbf{k}}$ is an SU(1,1) matrix.

Moreover, when the evolution consists of several steps $U(t) = U_n U_{n-1}...U_1$, we have:

$$
\begin{aligned}
\begin{pmatrix} a_{\mathbf{k}}(t) & a_{-\mathbf{k}}^\dagger(t) \end{pmatrix} &= U_1^\dagger...U_{n-1}^\dagger U_n^\dagger \begin{pmatrix} a_{\mathbf{k}} & a_{-\mathbf{k}}^\dagger \end{pmatrix} U_n U_{n-1}...U_1 \\
&= U_1^\dagger...U_{n-1}^\dagger \begin{pmatrix} a_{\mathbf{k}} & a_{-\mathbf{k}}^\dagger \end{pmatrix} U_{n-1}...U_1 \mathcal{U}_{\mathbf{k},n} \\
&= \begin{pmatrix} a_{\mathbf{k}} & a_{-\mathbf{k}}^\dagger \end{pmatrix} \mathcal{U}_{\mathbf{k},1}...\mathcal{U}_{\mathbf{k},n-1} \mathcal{U}_{\mathbf{k},n}.
\end{aligned}
\tag{8}
$$

That is to say, the full evolution $\mathcal{U}_{\mathbf{k}}(t) = \mathcal{U}_{\mathbf{k},1}...\mathcal{U}_{\mathbf{k},n-1}\mathcal{U}_{\mathbf{k},n}$.

In this paper, we focus on the periodic and quasi-periodic driving of the Bose-Einstein condensate $|\psi(t)\rangle = U(t)|\psi(0)\rangle$, and measure the total number of excitations $n_{\mathbf{k}}(t)$ in the final state. For concreteness, let us choose the initial state $|\psi(0)\rangle$ to be the fully condensed state with $a_{\mathbf{k}}|\psi(0)\rangle = 0$,[5] then after the evolution, we have

$$
n_{\mathbf{k}}(t) = \langle \psi(0)| U^\dagger(t) a_{\mathbf{k}}^\dagger a_{\mathbf{k}} U(t) |\psi(0)\rangle = |\beta_{\mathbf{k}}(t)|^2,
\tag{9}
$$

which is directly related to the matrix element of the SU(1,1) matrix $\mathcal{U}_{\mathbf{k}}(t)$. Besides $|\beta_{\mathbf{k}}(t)|^2$, one could also consider measuring other elements of $\mathcal{U}_{\mathbf{k}}$, by adding additional evolution right before the final measurement.

With the driving, particles in the condensate can be pumped into excitations with finite momentum. We call the system is in the heating phase at certain momentum $\mathbf{k}$ when the occupation number grows exponentially as $n_{\mathbf{k}}(t) \sim e^{\lambda_{\mathbf{k}} t}$ in the long-time limit, and we call the correspond exponent $\lambda_{\mathbf{k}} > 0$ the heating rate. From (9), this implies $|\beta_{\mathbf{k}}(t)|^2 \sim e^{\lambda_{\mathbf{k}} t}$, so does $|\alpha_{\mathbf{k}}(t)|^2 \sim e^{\lambda_{\mathbf{k}} t}$ due to the constraint $|\alpha_{\mathbf{k}}(t)|^2 - |\beta_{\mathbf{k}}(t)|^2 = 1$. Consequently, the kinetic energy of atoms with the corresponding momentum, which is defined as $k^2 n_{\mathbf{k}}/2$, grows exponentially in time, implying the system is being heated. In other cases, $n_{\mathbf{k}}$ may oscillates or increases polynomially, which we will refer to as the non-heating phase or the critical phase respectively.

As a side note, we would like to point out that the present discussion can be directly applied to specific dynamics of scale-invariant atomic gases in a trap [33,34]. To see the connection, let us consider a single harmonic oscillator[6] with the trapping frequency $\omega = 1$ and $a = \frac{1}{\sqrt{2}}(x+ip)$. Then we can define the following set of generators that satisfies the aforementioned $\mathfrak{su}(1,1)$ algebra:

$$
A^z = \frac{1}{4}\left(a^\dagger a + a a^\dagger\right), \qquad A^x = \frac{1}{4}\left((a^\dagger)^2 + a^2\right), \qquad A^y = \frac{1}{4i}\left((a^\dagger)^2 - a^2\right).
\tag{10}
$$

Consequently, for Hamiltonians given by a linear superposition of these operators, the dynamics can again be studied using the SU(1,1) group transformation. In the original basis, we can regroup the above generators into $p^2$, $x^2$ and $xp + px$. The generalization to many-body scale-invariant atomic gases, e.g. the unitary Fermi gas, can then be accomplished by noticing that these operators are a subgroup of the non-relativistic conformal group [41], where $p^2$, $x^2$ and $xp + px$ are generalized to time translation, special conformal transformation, and dilation.

---

[5]Note that this choice of the initial state is not essential for the classification of heating/non-heating phases, since the long-time exponential growth of the particle number only comes from the growth of $|\alpha_{\mathbf{k}}(t)|^2$ and $|\beta_{\mathbf{k}}(t)|^2$.

[6]We thank Ruihua Fan for pointing out to us the SU(1,1) dynamics of a single harmonic oscillator.

# 3 Periodic driving

In this section, we consider periodic drivings where $g(t) = g(t+T)$. We will discuss the phase diagram that is experimentally relevant. Another purpose for this section is to set up a base for the discussion in the quasi-periodic case.

Let us assume the total evolution time $t = nT$ with integer $n$, we have

$$U(nT) = U(T)^n, \qquad U(T) = \mathcal{T} \exp\left(-i \int_0^T H(t')dt'\right) = \exp(-iH_F T). \qquad (11)$$

Here $H_F$ is the standard Floquet Hamiltonian. From the perspective of the Heisenberg evolution of excitation operator (8), we have correspondingly $\mathcal{U}_{\mathbf{k}}(nT) = \mathcal{U}_{\mathbf{k}}(T)^n$. We are interested in the large $n$ asymptotics of $\mathcal{U}_{\mathbf{k}}(nT)$, which determines the dynamical phases of the condensate, namely the heating phase, the non-heating phase and the phase boundary (critical phase) between them.

The determination of the phase diagram has two steps.

1. We compute $\mathcal{U}_{\mathbf{k}}(T)$ for a given protocol specified by $g(t)$:

   The Heisenberg equation for $a_{\mathbf{k}}$ and $a_{-\mathbf{k}}^\dagger$ is given as follows

$$
\begin{aligned}
\frac{d\left(a_{\mathbf{k}}(t) \quad a_{-\mathbf{k}}^\dagger(t)\right)}{dt} &= \left(i[H(t), a_{\mathbf{k}}(t)] \quad i[H(t), a_{-\mathbf{k}}^\dagger(t)]\right) \\
&= -i\left(a_{\mathbf{k}}(t) \quad a_{-\mathbf{k}}^\dagger(t)\right) \underbrace{\begin{pmatrix} \frac{k^2}{2} + g(t)n_0 & -g(t)n_0 \\ g(t)n_0 & -\frac{k^2}{2} - g(t)n_0 \end{pmatrix}}_{=: \mathcal{M}_{\mathbf{k}}(t)}
\end{aligned}
\qquad (12)
$$

   or equivalently,

$$d\mathcal{U}_{\mathbf{k}}(t)/dt = -i\mathcal{U}_{\mathbf{k}}(t)\mathcal{M}_{\mathbf{k}}(t), \qquad \mathcal{U}_{\mathbf{k}}(T) = \widetilde{\mathcal{T}} \exp\left(-i \int_0^T \mathcal{M}_{\mathbf{k}}(t)dt\right), \qquad (13)$$

   with anti-time ordering operator $\widetilde{\mathcal{T}}$. In addition to performing the exact evaluation of $\mathcal{U}_{\mathbf{k}}(T)$, several standard approximations such as Magnus expansion [17] can also be applied, as discussed later in this section.

2. Analyze the large $n$ behavior of $\mathcal{U}_{\mathbf{k}}(T)^n$ based on a single $\mathcal{U}_{\mathbf{k}}(T)$.

   Assuming we have worked out

$$\mathcal{U}_{\mathbf{k}}(T) = \begin{pmatrix} \alpha_{\mathbf{k}}(T) & \beta_{\mathbf{k}}(T) \\ \beta_{\mathbf{k}}^*(T) & \alpha_{\mathbf{k}}^*(T) \end{pmatrix}. \qquad (14)$$

   Then the matrix element $\alpha_{\mathbf{k}}(nT)$ and $\beta_{\mathbf{k}}(nT)$ of $\mathcal{U}_{\mathbf{k}}(nT)$ can be related to the $\alpha_{\mathbf{k}}(T)$ and $\beta_{\mathbf{k}}(T)$ via following formulas

$$\left(\alpha_{\mathbf{k}}(nT), \ \beta_{\mathbf{k}}(nT)\right) = \begin{cases} \left(\frac{\eta_{\mathbf{k}}^{-\frac{n}{2}}\gamma_{\mathbf{k},1} - \eta_{\mathbf{k}}^{\frac{n}{2}}\gamma_{\mathbf{k},2}}{\gamma_{\mathbf{k},1} - \gamma_{\mathbf{k},2}}, \ \frac{(\eta_{\mathbf{k}}^{-\frac{n}{2}} - \eta_{\mathbf{k}}^{\frac{n}{2}})\gamma_{\mathbf{k},1}\gamma_{\mathbf{k},2}}{\gamma_{\mathbf{k},1} - \gamma_{\mathbf{k},2}}\right) & \text{if } |\mathrm{Tr}(\mathcal{U}_{\mathbf{k}}(T))| \neq 2 \\ \left(1 + n\gamma_{\mathbf{k}}\beta_{\mathbf{k}}^*(T), \ -n\gamma_{\mathbf{k}}^2\beta_{\mathbf{k}}^*(T)\right) & \text{if } |\mathrm{Tr}(\mathcal{U}_{\mathbf{k}}(T))| = 2 \end{cases}. \qquad (15)$$

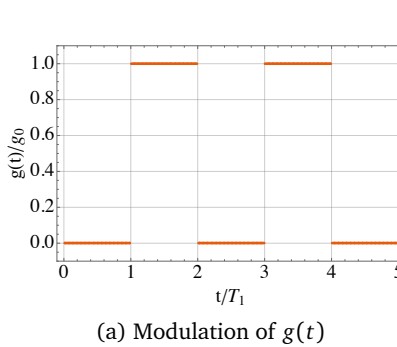

(a) Modulation of $g(t)$

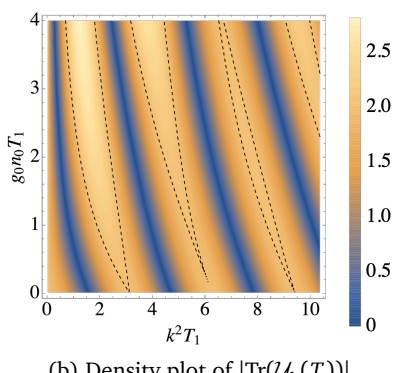

(b) Density plot of $|\mathrm{Tr}(\mathcal{U}_\mathbf{k}(T))|$

Figure 1: (a). The square-wave periodic modulation of the scattering length. Here we choose the special case with $T_0 = T_1$ and $g_1 = 0$. (b). A density plot of $|\mathrm{Tr}(\mathcal{U}_\mathbf{k}(T))|$ for $T_0 = T_1$ and $g_1 = 0$. This determines $\lambda_\mathbf{k}$ for $|\mathrm{Tr}(\mathcal{U}_\mathbf{k}(T))| > 2$ (heating phase) or $\omega_\mathbf{k}$ for $|\mathrm{Tr}(\mathcal{U}_\mathbf{k}(T))| < 2$ (non-heating phase). The dashed line denotes the phase boundary, where $|\mathrm{Tr}(\mathcal{U}_\mathbf{k}(T))| = 2$.

Here we have defined

$$
\gamma_{\mathbf{k},1(2)} = \frac{1}{2\beta_\mathbf{k}(T)}\left[\alpha_\mathbf{k}(T) - \alpha_\mathbf{k}^*(T) \mp \sqrt{\mathrm{Tr}(\mathcal{U}_\mathbf{k}(T))^2 - 4}\right],
$$

$$
\eta_\mathbf{k} = \frac{\mathrm{Tr}(\mathcal{U}_\mathbf{k}(T)) + \sqrt{\mathrm{Tr}(\mathcal{U}_\mathbf{k}(T))^2 - 4}}{\mathrm{Tr}(\mathcal{U}_\mathbf{k}(T)) - \sqrt{\mathrm{Tr}(\mathcal{U}_\mathbf{k}(T))^2 - 4}}. \tag{16}
$$

Note $\gamma_{\mathbf{k},1} = \gamma_{\mathbf{k},2} = \gamma_\mathbf{k}$ for $|\mathrm{Tr}(\mathcal{U}_\mathbf{k}(T))| = 2$.

From (15), we conclude that the $\beta_\mathbf{k}(nT)$ grows exponentially only when $|\mathrm{Tr}(\mathcal{U}_\mathbf{k}(T))| > 2$, and the exponent reads

$$
\lambda_\mathbf{k} = \frac{2}{T}\log\left(\frac{|\mathrm{Tr}(\mathcal{U}_\mathbf{k}(T))| + \sqrt{\mathrm{Tr}(\mathcal{U}_\mathbf{k}(T))^2 - 4}}{2}\right). \tag{17}
$$

When $|\mathrm{Tr}(\mathcal{U}_\mathbf{k}(T))| < 2$, the $\eta$ becomes a pure phase and $n_\mathbf{k}(nT)$ oscillates periodically

$$
n_\mathbf{k}(nT) = n_\mathbf{k}(T)\frac{\sin^2 \delta_\mathbf{k} n}{\sin^2 \delta_\mathbf{k}}, \qquad \delta_\mathbf{k} = \arg\left[|\mathrm{Tr}(\mathcal{U}_\mathbf{k}(T))| + i\sqrt{4 - \mathrm{Tr}(\mathcal{U}_\mathbf{k}(T))^2}\right], \tag{18}
$$

with frequency $\omega_\mathbf{k} = 2|\delta_\mathbf{k}|/T$. When $\delta_\mathbf{k}/\pi = p/q$ is a rational number, we have $n_\mathbf{k}(nqT) = 0$. At the phase boundary with $|\mathrm{Tr}(\mathcal{U}_\mathbf{k}(T))| = 2$, $n_\mathbf{k}(nT)$ grows quadratically if we have $\beta_\mathbf{k}(T) \neq 0$.

In the following, we consider two concrete protocols, and determine the phase diagram in terms of experimental parameters.

## 3.1 Protocol 1: square-wave

In this protocol, the scattering length is tuned to be a square wave as shown in Figure 1(a). We then have $U(T) = U_0(T_0)U_1(T_1) = e^{-iH_0 T_0}e^{-iH_1 T_1}$, where in $H_{j=0,1}$ the interaction strength is $g_j$. To avoid instability, we assume both $g_j \geqslant 0$. Using (12), we have

$$
\mathcal{U}_{\mathbf{k},j}(T_j) = \begin{pmatrix} \cos(E_{\mathbf{k},j}T_j) - \frac{iA_{\mathbf{k},j}}{E_{\mathbf{k},j}}\sin(E_{\mathbf{k},j}T_j) & \frac{iB_j}{E_{\mathbf{k},j}}\sin(E_{\mathbf{k},j}T_j) \\ -\frac{iB_j}{E_{\mathbf{k},j}}\sin(E_{\mathbf{k},j}T_j) & \cos(E_{\mathbf{k},j}T_j) + \frac{iA_{\mathbf{k},j}}{E_{\mathbf{k},j}}\sin(E_{\mathbf{k},j}T_j) \end{pmatrix}, \tag{19}
$$

where $A_{\mathbf{k},j} = \frac{k^2}{2} + g_j n_0$, $B_j = g_j n_0$ and $E_{\mathbf{k},j} = \sqrt{A_{\mathbf{k},j}^2 - B_j^2} = \sqrt{k^2 g_j n_0 + k^4/4}$ is the energy of phonons with corresponding interaction strength.

We then compute $\mathcal{U}_{\mathbf{k}}(T) = \mathcal{U}_{\mathbf{k},1}(T_1)\mathcal{U}_{\mathbf{k},0}(T_0)$ and find

$$\text{Tr}(\mathcal{U}_{\mathbf{k}}(T)) = 2\cos(E_{\mathbf{k},1}T_1)\cos(E_{\mathbf{k},0}T_0) + 2\sin(E_{\mathbf{k},1}T_1)\sin(E_{\mathbf{k},0}T_0)\frac{B_1 B_0 - A_{\mathbf{k},1}A_{\mathbf{k},0}}{E_{\mathbf{k},1}E_{\mathbf{k},0}}. \tag{20}$$

When $|\mathbf{k}|$ is large, we expect the effect of $g$ being small and no particles can be excited. This is consistent with (20), where we have $\text{Tr}(\mathcal{U}_{\mathbf{k}}(T)) \approx 2\cos(k^2(T_0 + T_1)/2) \leqslant 2$ and $\beta_{\mathbf{k}}(T) \approx 0$.

For general $|\mathbf{k}|$, we compute (20) numerically and plot $|\text{Tr}(\mathcal{U}_{\mathbf{k}}(T))|$ in Figure 1 (b) which can be served as a phase diagram. To be concrete, we fix $T_0 = T_1$ and $g_1 = 0$. At small $g_0$, the heating phase appears near the resonance $k^2 T = 2n\pi$, where 2 particles can be excited by the periodic modulation.

It is also interesting to draw an analogy between the square-wave driving protocol and a one dimensional tight-binding model with Hamiltonian

$$H = \sum_j \big( |j\rangle\langle j+1| + |j+1\rangle\langle j| + V_j |j\rangle\langle j| \big). \tag{21}$$

For a given energy $E$, the wave function satisfies $E\psi_j = \psi_{j-1} + \psi_{j+1} + V_j\psi_j$, with $\psi_j$ being the position-space wavefunction. Denoting $\Psi_j = (\psi_j, \psi_{j-1})^T$, we have $\Psi_{j+1} = \mathbf{T}_j\mathbf{T}_{j-1}...\mathbf{T}_1\Psi_1$ with the transfer matrix

$$\mathbf{T}_j = \begin{pmatrix} E - V_j & -1 \\ 1 & 0 \end{pmatrix}. \tag{22}$$

Similar to the driving $\mathcal{U}_{\mathbf{k},j}(T_j)$, the transfer matrix also has determinant 1, i.e. $\det(\mathbf{T}_j) = 1$. Now, given the initial wavefunction $\Psi_1$, the large $n$ asymptotics of wavefunction $\Psi_n$ is determined by the eigenvalues of transfer matrices. Therefore, analyzing the long-distance behavior of the wavefunction in tight-binding models is a direct analogy of analyzing the long time dynamics of the excitations in the driven BEC. More explicitly, when the energy $E$ belongs to a band, the wavefunction oscillates in space and it corresponds to the non-heating phase in driven BEC where the number of excitations is oscillating in time. On the other hand, if $E$ lies in a band-gap, the wavefunction grows exponentially, indicating no valid bulk state, and corresponds to the heating phase in driven BEC.

## 3.2 Protocol 2: sine-wave

In this protocol, we let $g(t)$ be a single or a combination of sine functions. We first consider the case where $g(t) = 2g_0\cos(\omega_0 t)$ and $\omega_0 = 2\pi/T$. Using (12), we further write out the set of equations satisfied by $\alpha_{\mathbf{k}}(t)$ and $\beta_{\mathbf{k}}(t)$:

$$\begin{aligned}
\frac{d\alpha_{\mathbf{k}}(t)}{dt} &= -i\left(\frac{k^2}{2} + g(t)n_0\right)\alpha_{\mathbf{k}}(t) - ig(t)n_0\beta_{\mathbf{k}}(t), \\
\frac{d\beta_{\mathbf{k}}(t)}{dt} &= i\left(\frac{k^2}{2} + g(t)n_0\right)\beta_{\mathbf{k}}(t) + ig(t)n_0\alpha_{\mathbf{k}}(t).
\end{aligned} \tag{23}$$

This leads to [18]

$$\frac{d^2(\alpha_{\mathbf{k}}(t) - \beta_{\mathbf{k}}(t))}{dt^2} + \frac{k^2}{2}\left(\frac{k^2}{2} + 2g(t)n_0\right)(\alpha_{\mathbf{k}}(t) - \beta_{\mathbf{k}}(t)) = 0. \tag{24}$$

Regarding $t$ as the spatial dimension, this is the Schrödinger equation for a particle in 1D with mass $m = 1/2$ and energy $E = \frac{k^4}{4}$, moving in a potential with $V(t) = -2k^2 n_0 g_0\cos(\omega_0 t)$.

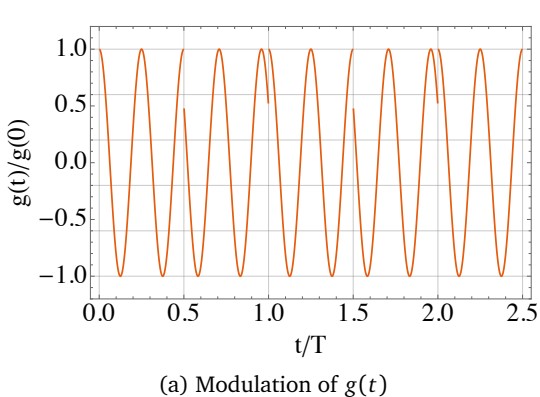

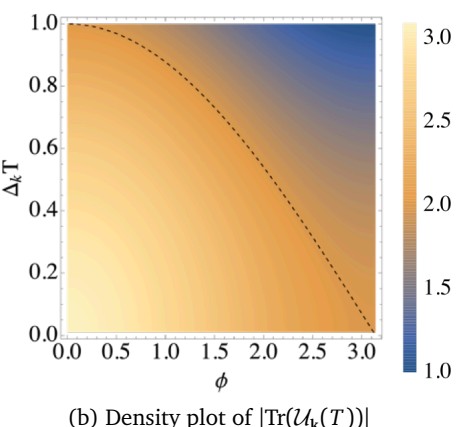

(a) Modulation of $g(t)$        (b) Density plot of $|\text{Tr}(\mathcal{U}_{\mathbf{k}}(T))|$

Figure 2: (a). The sine-wave periodic modulation of the scattering length. Here we set $\phi = \pi/3$. (b). A density plot of $|\text{Tr}(\mathcal{U}_{\mathbf{k}}(T))|$ with $g_0 n_0 T = 1$. This determines $\lambda_{\mathbf{k}}$ for $|\text{Tr}(\mathcal{U}_{\mathbf{k}}(T))| > 2$ (heating phase) and $\omega_{\mathbf{k}}$ for $|\text{Tr}(\mathcal{U}_{\mathbf{k}}(T))| < 2$ (non-heating phase). The dashed line denotes the phase boundary, where $|\text{Tr}(\mathcal{U}_{\mathbf{k}}(T))| = 2$.

Similar to the discussion in the last section, the system would be in the non-heating phase if the corresponding energy lies in some bands and otherwise, it will be in the heating phase. The detailed phase diagram has been worked out in [18]. Here we instead focus on the case with large $\omega_0 \gg g_0 n_0$ and near resonance $\omega_0 - k^2 \ll \omega_0$, where we could perform a rotation-wave approximation to (12) and find

$$R_{\mathbf{k}}(t) = e^{-ik^2 \sigma_z/2}, \qquad \widetilde{\mathcal{M}_{\mathbf{k}}} = \overline{R_{\mathbf{k}}(t)\mathcal{M}_{\mathbf{k}}(t)R_{\mathbf{k}}^{\dagger}(t) + iR_{\mathbf{k}}(t)\partial_t R_{\mathbf{k}}^{\dagger}(t)} = \begin{pmatrix} \Delta_{\mathbf{k}} & -g_0 n_0 \\ g_0 n_0 & -\Delta_{\mathbf{k}} \end{pmatrix}. \quad (25)$$

Here $\Delta_{\mathbf{k}} = k^2/2 - \omega_0/2$ and the average is performed over time, i.e. $\overline{O(t)} := \frac{1}{T}\int_0^T O(t)dt$. Systematic improvements can be made by taking into account the $1/\omega_0$ correction. In the rotating frame, $\mathcal{U}_{\mathbf{k}}(T)$ now has the same form as (19), with $A_{\mathbf{k}} = \Delta_{\mathbf{k}}$, $B = g_0 n_0$ and $E_{\mathbf{k}} = \sqrt{\Delta_{\mathbf{k}}^2 - g_0^2 n_0^2}$, leading to:

$$\text{Tr}(\mathcal{U}_{\mathbf{k}}(T)) = 2\cos\left(T\sqrt{\Delta_{\mathbf{k}}^2 - g_0^2 n_0^2}\right). \quad (26)$$

When $\Delta_{\mathbf{k}} < g_0 n_0$, the system is in the heating phase, with a heating rate $\lambda_{\mathbf{k}} = \sqrt{\Delta_{\mathbf{k}}^2 - g_0^2 n_0^2}$. On the other hand, for $\Delta_{\mathbf{k}} > g_0 n_0$, the system is in the non-heating phase, with $\omega_{\mathbf{k}} = \sqrt{g_0^2 n_0^2 - \Delta_{\mathbf{k}}^2}$.

We then consider a combined protocol where we divide each period into two halves as in the square wave case, as shown in Figure 2 (a). In the first half period, we modulate the scattering length as $g(t \in [0, T/2)) = 2g_0 \cos(\omega_0 t)$ and in the second period we add an additional phase shift $g(t \in [T/2, T)) = 2g_0 \cos(\omega_0 t + \phi)$. We assume $T = 4\pi n/\omega_0$, with $n \in \mathbb{Z}$. Due to the additional phase, we now have

$$\widetilde{\mathcal{M}_{\mathbf{k},1}} = \begin{pmatrix} \Delta_{\mathbf{k}} & -g_0 n_0 \\ g_0 n_0 & -\Delta_{\mathbf{k}} \end{pmatrix}, \qquad \widetilde{\mathcal{M}_{\mathbf{k},2}} = \begin{pmatrix} \Delta_{\mathbf{k}} & -g_0 n_0 e^{i\phi} \\ g_0 n_0 e^{-i\phi} & -\Delta_{\mathbf{k}} \end{pmatrix}. \quad (27)$$

Therefore the trace of $\mathcal{U}_{\mathbf{k}}(T) = \exp(-i\widetilde{\mathcal{M}_{\mathbf{k},1}}T/2)\exp(-i\widetilde{\mathcal{M}_{\mathbf{k},2}}T/2)$ is expressed as follows,

$$\text{Tr}(\mathcal{U}_{\mathbf{k}}(T)) = \frac{-2B^2 \sin^2 \frac{\phi}{2} + \cos(E_{\mathbf{k}}T)(2\Delta_{\mathbf{k}}^2 - 2B^2 \cos^2 \frac{\phi}{2})}{E_{\mathbf{k}}^2}. \quad (28)$$

We plot the numerical result of $|\text{Tr}(\mathcal{U}_{\mathbf{k}}(T))|$ in Figure 2 (b). For $\Delta_{\mathbf{k}} = 0$, the system transits into the non-heating phase at $\phi = \pi$ for large $\omega_0$, as observed in the experiment [15]. Note that at this point, the second half of the evolution cancels exactly with the first half, but this exact cancellation is not the general picture for a generic phase boundary ($\text{Tr}(\mathcal{U}_{\mathbf{k}}(T)) = 2$). When including $1/\omega_0$ corrections, the phase boundary would shift, as studied in [17].

# 4 Quasi-periodic driving

Now we turn to the quasi-periodically driven condensate where $g(t)$ is deterministic but has no periodicity. We consider both square-wave and sine-wave protocols as in the periodic driving case. The square-wave protocol is a time-domain analog of the Fibonacci model for the one-dimensional quasi-crystal (but generalizing to an arbitrary irrational number following [42, 43]), while the sine-wave protocol corresponds to the quasi-periodic Mathieu's equation, whose lattice version is also known as the Audry-André model [44].

## 4.1 Protocol 1: square-wave

### 4.1.1 The set-up and the trace map

For this square-wave quasi-periodicity protocol, we define a sequence of 0/1 bits with an irrational number $\alpha \in (0,1)$

$$V_j = \lfloor (j+1)\alpha \rfloor - \lfloor j\alpha \rfloor. \tag{29}$$

Here $\lfloor y \rfloor$ is the floor function which gives the greatest integer less than or equal to $y$.[7] For example, for the inverse golden ratio $\alpha = \frac{\sqrt{5}-1}{2}$, we have

$$V_{1,2,3\ldots} = 10110101\ldots, \tag{31}$$

which is also known as Fibonacci word.

Next, we evolve the system according to the sequence: if the bit is "1/0", we apply the unitary $U_{1/0}$ respectively, with $U_1 = \mathcal{T}\exp(-i\int_0^T dt H_1(t))$ and $U_0 = \mathcal{T}\exp(-i\int_0^T dt H_0(t))$. Again, for the inverse golden ratio example, we have

$$U(T) = U_1, \quad U(2T) = U_0 U_1, \quad U(3T) = U_1 U_0 U_1, \quad U(5T) = U_0 U_1 U_1 U_0 U_1, \quad \ldots \tag{32}$$

or in terms of $\mathcal{U}_{\mathbf{k},1}$ and $\mathcal{U}_{\mathbf{k},0}$:

$$
\begin{aligned}
\mathcal{U}_{\mathbf{k}}(T) &= \mathcal{U}_{\mathbf{k},1}, \quad \mathcal{U}_{\mathbf{k}}(2T) = \mathcal{U}_{\mathbf{k},1}\mathcal{U}_{\mathbf{k},0}, \quad \mathcal{U}_{\mathbf{k}}(3T) = \mathcal{U}_{\mathbf{k},1}\mathcal{U}_{\mathbf{k},0}\mathcal{U}_{\mathbf{k},1}, \\
\mathcal{U}_{\mathbf{k}}(5T) &= \mathcal{U}_{\mathbf{k},1}\mathcal{U}_{\mathbf{k},0}\mathcal{U}_{\mathbf{k},1}\mathcal{U}_{\mathbf{k},1}\mathcal{U}_{\mathbf{k},0}, \quad \ldots
\end{aligned}
\tag{33}
$$

Note the order is reversed according to the definition (8).

The evolution can be described more efficiently if we only probe the system at specific time. This simplification relies on the continued fraction representation for $\alpha$:

$$\alpha = a_0 + \cfrac{1}{a_1 + \cfrac{1}{a_2 + \ldots \frac{1}{a_n + \ldots}}}, \tag{34}$$

---

[7] This definition is equivalent to the other definition that is commonly used in the quasi-crystal literature

$$V(j) = \chi_{[1-\alpha,1)}(j\alpha), \tag{30}$$

where $\chi_{[1-\alpha,1)}(t) = \chi_{[1-\alpha,1)}(t+1)$ is a periodic characteristic function with period 1, namely $\chi_{[1-\alpha,1)}(t) = 1$ if $1-\alpha \leqslant t < 1$, and $\chi_{[1-\alpha,1)}(t) = 0$ if $0 \leqslant t < 1-\alpha$. See e.g. [45, 46] for the Fibonacci model when $\alpha$ is the inverse golden ratio.

where $\{a_n\}_{n=1,2,3...}$ are a sequence of positive integers, and $a_0 = \lfloor \alpha \rfloor$ could be negative or 0, for our case $\alpha \in (0,1)$, we have $a_0 = 0$. Any real number has an unique continued fraction representation, for rational numbers, the continued fractions terminate at finite $n$, while the irrational numbers have infinite sequences $[a_0, a_1, a_2 \ldots]$.

Continued fractions are useful in finding the best Diophantine approximations. Operationally, we can truncate the continued fraction at order $n$, which produces a rational number (known as $n$-th principal convergent)

$$\frac{p_n}{q_n} = a_0 + \cfrac{1}{a_1 + \cfrac{1}{a_2 + \ldots \frac{1}{a_n}}}, \tag{35}$$

where integers $(p_n, q_n)$ satisfies the recursion relation $q_n = a_n q_{n-1} + q_{n-2}$ and $p_n = a_n p_{n-1} + p_{n-2}$. For example, for the inverse golden ratio $\alpha = \frac{\sqrt{5}-1}{2}$, we have $a_0 = 0$, $a_{n>0} = 1$ and $q_n$ are given by the Fibonacci numbers: $q_1 = 1$, $q_2 = 2$ and $q_n = q_{n-1} + q_{n-2}$.

These rational numbers $p_n/q_n$ provide best rational approximations of the irrational $\alpha$ in the sense that the product $q_n \alpha$ is closer to an integer than any smaller $q < q_n$ (see e.g. [47] for more explanations). Consequently, one can show that[8]

$$V_{j+q_{n-1}} = V_j, \qquad 1 \leqslant j < q_n - 1, \tag{36}$$

for $V_j$ defined in (29). This relation says the first $q_n$ elements of the sequence can be determined by copying the first $q_{n-1}$ elements. Thus, we have the following recursion relation

$$\mathcal{U}_{\mathbf{k}}(q_n T) = \mathcal{U}_{\mathbf{k}}(q_{n-1} T)^{a_n} \mathcal{U}_{\mathbf{k}}(q_{n-2} T). \tag{37}$$

An important analytic tool that allows us to efficiently determine the heating rate is the trace map [42, 45], which is a recursion relation among the following traces

$$x_{\mathbf{k}}^n = \frac{1}{2} \text{Tr} \left( \mathcal{U}_{\mathbf{k}}(q_n T) \mathcal{U}_{\mathbf{k}}(q_{n-1} T) \right), \quad y_{\mathbf{k}}^n = \frac{1}{2} \text{Tr} \left( \mathcal{U}_{\mathbf{k}}(q_n T) \right), \quad z_{\mathbf{k}}^n = \frac{1}{2} \text{Tr} \left( \mathcal{U}_{\mathbf{k}}(q_{n-1} T) \right). \tag{38}$$

The trace map is given as follows,

$$\begin{aligned} x_{\mathbf{k}}^n &= S^{a_n}(y_{\mathbf{k}}^{n-1}) x_{\mathbf{k}}^{n-1} - S^{a_n - 1}(y_{\mathbf{k}}^{n-1}) z_{\mathbf{k}}^{n-1}, \\ y_{\mathbf{k}}^n &= S^{a_n - 1}(y_{\mathbf{k}}^{n-1}) x_{\mathbf{k}}^{n-1} - S^{a_n - 2}(y_{\mathbf{k}}^{n-1}) z_{\mathbf{k}}^{n-1}, \\ z_{\mathbf{k}}^n &= y_{\mathbf{k}}^{n-1}, \end{aligned} \tag{39}$$

where $S^n(y) = \frac{\sin[(n+1)\arccos(y)]}{\sin[\arccos(y)]}$ is the Chebyshev polynomial of second kind (see appendix for a derivation). Using the trace map (39), one can straightforwardly show that

$$I = (x_{\mathbf{k}}^n)^2 + (y_{\mathbf{k}}^n)^2 + (z_{\mathbf{k}}^n)^2 - 2 x_{\mathbf{k}}^n y_{\mathbf{k}}^n z_{\mathbf{k}}^n - 1, \tag{40}$$

is independent of $n$, hence an integral of motion (Fricke-Vogt invariant). When $I > 0$, for almost all initial conditions, $(x_{\mathbf{k}}^n, y_{\mathbf{k}}^n, z_{\mathbf{k}}^n)$ flows to infinity as $n$ increases.

Now let us consider explicit examples: we choose $H_1(t)$ to be the non-interacting Hamiltonian with $g(t) = 0$ and $H_0(t)$ to be the Hamiltonian with constant interaction $g_0 > 0$. Using (19), we have

$$\mathcal{U}_{\mathbf{k},1} = \begin{pmatrix} e^{-i\epsilon_{\mathbf{k}} T} & 0 \\ 0 & e^{i\epsilon_{\mathbf{k}} T} \end{pmatrix}, \quad \mathcal{U}_{\mathbf{k},0} = \begin{pmatrix} \cos(E_{\mathbf{k}} T) - \frac{iA_{\mathbf{k}}}{E_{\mathbf{k}}} \sin(E_{\mathbf{k}} T) & \frac{iB}{E_{\mathbf{k}}} \sin(E_{\mathbf{k}} T) \\ -\frac{iB}{E_{\mathbf{k}}} \sin(E_{\mathbf{k}} T) & \cos(E_{\mathbf{k}} T) + \frac{iA_{\mathbf{k}}}{E_{\mathbf{k}}} \sin(E_{\mathbf{k}} T) \end{pmatrix}. \tag{41}$$

---

[8]For details of the proof, see e.g. [43]

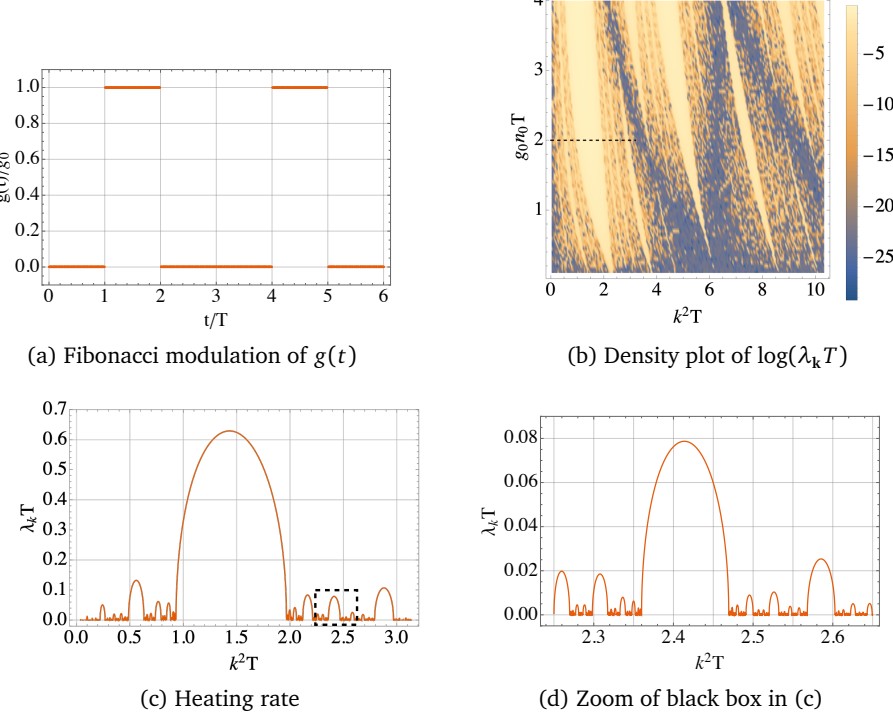

(a) Fibonacci modulation of $g(t)$

(b) Density plot of $\log(\lambda_{\mathbf{k}}T)$

(c) Heating rate

(d) Zoom of black box in (c)

Figure 3: (a). The Fibonacci modulation of the scattering length. (b). A density plot of $\log(\lambda_{\mathbf{k}}T)$ determined by taking a period of $q_{50} \sim 2 \times 10^{10}$ steps. The system is in the heating phase almost everywhere. (c-d). The the heating rate $\lambda_{\mathbf{k}}T$ with fixed $g_0 n_0 T = 2$ determined by taking a period of $q_{50} \sim 2 \times 10^{10}$ steps. The results show self-similar structures as in [30].

Recall that we have $\epsilon_{\mathbf{k}} = k^2/2$, $A_{\mathbf{k}} = \epsilon_{\mathbf{k}} + g_0 n_0$, $B = g_0 n_0$ and $E_{\mathbf{k}} = \sqrt{A_{\mathbf{k}}^2 - B^2}$.

In this protocol, the heating rate $\lambda_{\mathbf{k}}$ can be approximated by taking the large $n$ limit of periodic driving with a period $q_n T$: we could approximate $\alpha \approx p_n/q_n$ and use (29) to define the driving sequence within a period. Following (17), this gives

$$\lambda_{\mathbf{k}}(q_n) = \frac{2}{q_n T} \log\left(|y_{\mathbf{k}}^n| + \sqrt{(y_{\mathbf{k}}^n)^2 - 1}\right), \tag{42}$$

with sufficiently large $n$. Generally $q_n$ grows exponentially with $n$ and we can obtain accurate results for $n$ being a few tenths.

### 4.1.2 Patterns in phase diagram

To proceed, we also need to choose the irrational number $\alpha$, namely choose the modulation pattern that is determined by $V_j$ sequence. We will exam two specific examples in this subsection (1) $\alpha = \frac{\sqrt{5}-1}{2}$, which is the most popular choice in the study of 1D quasi-crystal and has the name "Fibonacci model" [45,46]; (2) $\alpha = \pi - 3$.

**(1) Fibonacci Driving.** Let us start with the inverse golden ratio $\alpha = \frac{\sqrt{5}-1}{2}$, which is called the Fibonacci driving in [30]. In this case, since $a_{n>0} = 1$, we have

$$\mathcal{U}_{\mathbf{k}}(q_n T) = \mathcal{U}_{\mathbf{k}}(q_{n-1} T)\mathcal{U}_{\mathbf{k}}(q_{n-2} T), \qquad x_{\mathbf{k}}^n = y_{\mathbf{k}}^{n+1}. \tag{43}$$

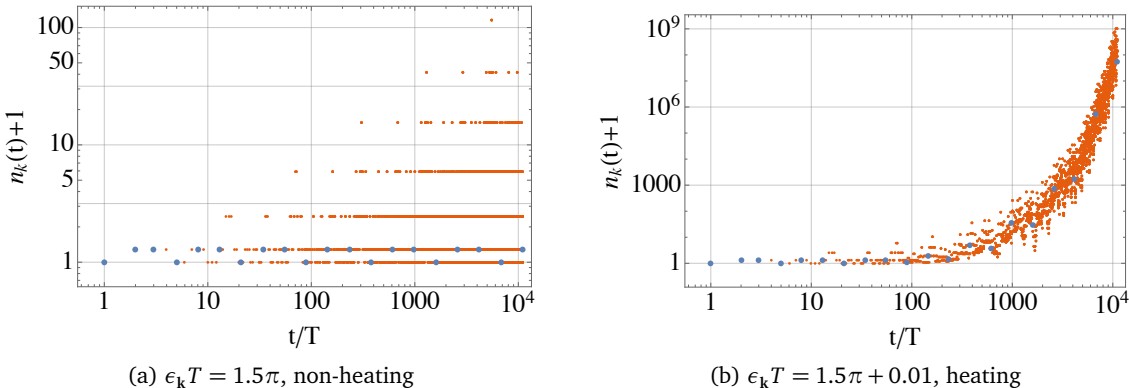

(a) $\epsilon_{\mathbf{k}} T = 1.5\pi$, non-heating  (b) $\epsilon_{\mathbf{k}} T = 1.5\pi + 0.01$, heating

Figure 4: The Fibonacci modulation of the scattering length: (a). $n_{\mathbf{k}}(t)$ as a function of $t/T$ for $\epsilon_{\mathbf{k}} T = 1.5\pi$ and $E_{\mathbf{k}} T = 2.5\pi$ without exponential heating. (b). $n_{\mathbf{k}}(t)$ as a function of $t/T$ for $\epsilon_{\mathbf{k}} T = 1.5\pi + 0.01$ and $E_{\mathbf{k}} T = 2.5\pi$ with exponential heating. The blue dots are the results at Fibonacci numbers $t = q_n T$.

One can extend the recursion relation (39) to $n = 2$ by defining $\mathcal{U}_{\mathbf{k}}(q_0 T) \equiv \mathcal{U}_{\mathbf{k},0}$, which gives

$$
\begin{aligned}
x_{\mathbf{k}}(F_0) &= \cos(E_{\mathbf{k}} T), \quad x_{\mathbf{k}}(F_1) = \cos(\epsilon_{\mathbf{k}} T), \\
x_{\mathbf{k}}(F_2) &= \cos(E_{\mathbf{k}} T)\cos(\epsilon_{\mathbf{k}} T) - \sin(E_{\mathbf{k}} T)\sin(\epsilon_{\mathbf{k}} T)\frac{A_{\mathbf{k}}}{E_{\mathbf{k}}},
\end{aligned}
\tag{44}
$$

and consequently

$$
I = \frac{g_0^2 n_0^2}{E_{\mathbf{k}}^2}\sin^2(E_{\mathbf{k}} T)\sin^2(\epsilon_{\mathbf{k}} T).
\tag{45}
$$

The $I = 0$ case where either $\sin(E_{\mathbf{k}} T) = 0$ or $\sin(\epsilon_{\mathbf{k}} T) = 0$ corresponds to a single quench (shown as black dashed line in Fig. 6) and we should focus on the $I > 0$ case. It is known that when $I > 0$, the phase with zero heating rate $\lambda_{\mathbf{k}}$ forms a Cantor set of measure zero in the phase diagram [43] and the system would generally be heated up exponentially in time. This is consistent with the numerical results shown in Figure 3. As shown in (b), the heating rate is non-zero almost everywhere, although the magnitude can be arbitrarily small with a self-similar structure as shown in (c-d). Later we will comment, with a comparison to the $\pi$-driving, the self-similarity in the Fibonacci driving is a manifestation of the self-similarity in its continued fraction representation

$$
\frac{\sqrt{5}-1}{2} = \frac{1}{1 + \frac{1}{1 + \frac{1}{1 + \dots}}}.
\tag{46}
$$

Although the measure of phases with $\lambda_{\mathbf{k}} = 0$ is zero, special points with $\lambda_{\mathbf{k}} = 0$ are known and interesting. When two of the initial conditions $(x_{\mathbf{k}}^n, y_{\mathbf{k}}^n, z_{\mathbf{k}}^n)$ become zero, $y_{\mathbf{k}}^n$ is then a periodic function of $n$ with a period of 6. As an example, for $(x_{\mathbf{k}}^1, y_{\mathbf{k}}^1, z_{\mathbf{k}}^1) = (a, 0, 0)$ we have

$$
y_{\mathbf{k}}^{6n} = y_{\mathbf{k}}^{6n+1} = y_{\mathbf{k}}^{6n+3} = y_{\mathbf{k}}^{6n+4} = 0, \qquad y_{\mathbf{k}}^{6n+3} = a, \qquad y_{\mathbf{k}}^{6n+5} = -a.
\tag{47}
$$

It is straightforward to show that the number of excited particles oscillates periodically with respect to n at time $q_n T$ and the period is 3. On the other hand, if one probes the system at non-Fibonacci number times, there are still excitations and the number of excitations grows polynomially. An explicit example is shown in Fig. 4 (a) (with $\epsilon_{\mathbf{k}} T = 1.5\pi$ and $E_{\mathbf{k}} T = 2.5\pi$) compared to the exponentially heating (b) if we slightly perturb away from the special points (with $\epsilon_{\mathbf{k}} T = 1.5\pi + 0.01$ and $E_{\mathbf{k}} T = 2.5\pi$).

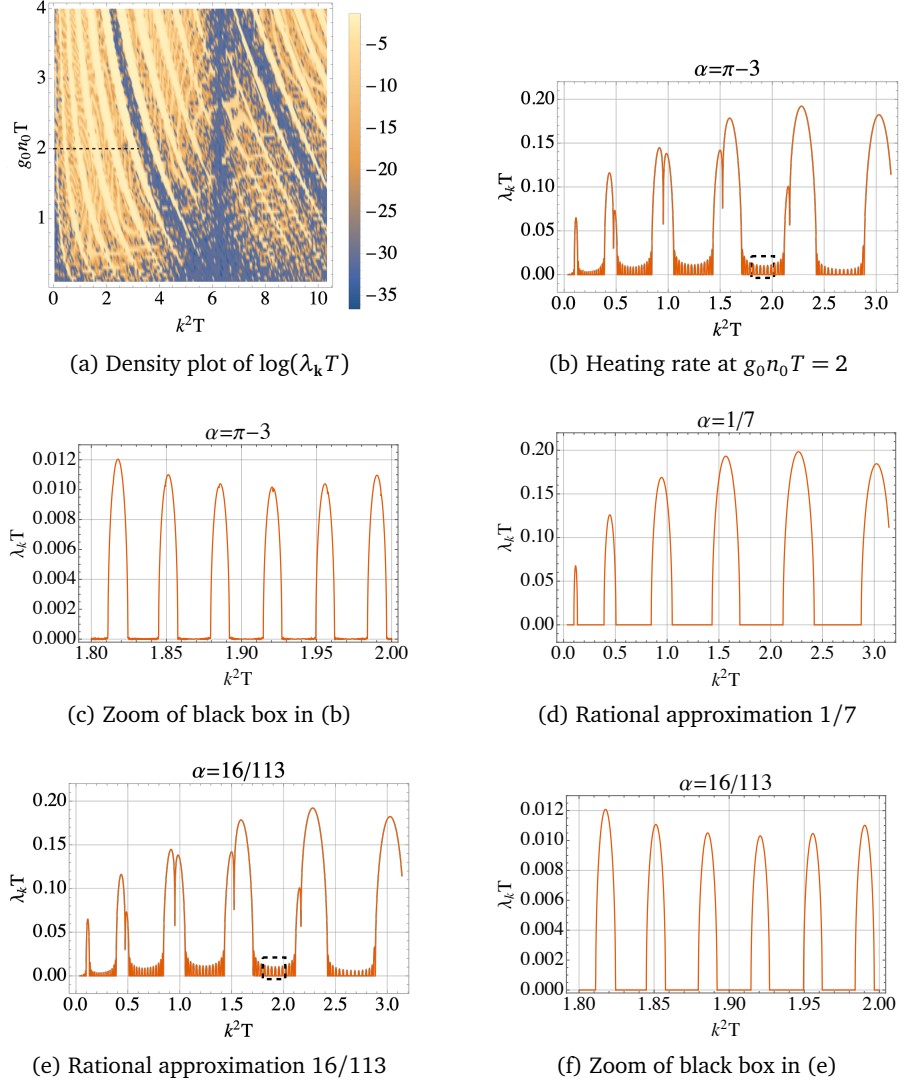

(a) Density plot of $\log(\lambda_{\mathbf{k}}T)$

(b) Heating rate at $g_0 n_0 T = 2$

(c) Zoom of black box in (b)

(d) Rational approximation $1/7$

(e) Rational approximation $16/113$

(f) Zoom of black box in (e)

Figure 5: Results for the $\pi$ modulation of the scattering length. (a). A density plot of $\log(\lambda_{\mathbf{k}}T)$ determined by taking a period of $q_{25} \sim 4 \times 10^{13}$ steps. (b-c). The 1D plot for the heating rate $\lambda_{\mathbf{k}}T$ with fixed $g_0 n_0 T = 2$. (d). The heating rate for rational approximation $x = p_1/q_1 = 1/7$. (e-f). The heating rate for rational approximation $x = p_3/q_3 = 16/113$. By comparing (b-f), we find $p_1/q_1$ and $p_3/q_3$ governs the heating rate at different magnitude scales.

**(2) $\pi$ Driving.** Now we consider the choice $\alpha = \pi - 3 \in (0,1)$, the point is that such an irrational number has a rather irregular continued fraction representation, unlike the case for the inverse gold ratio, whose continued fraction representation has self-similar pattern. We will numerically show that the patterns in the phase diagram reassemble the patterns in the continued fraction of $\pi - 3$:

$$a_1 = 7, \quad a_2 = 15, \quad a_3 = 1, \quad a_4 = 292 \quad \dots \tag{48}$$

that is to say

$$\pi - 3 = \cfrac{1}{7 + \cfrac{1}{15 + \cfrac{1}{1 + \cfrac{1}{292 + \dots}}}}, \tag{49}$$

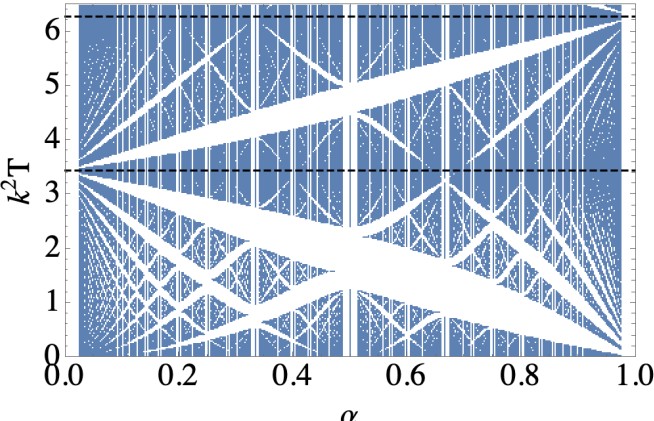

Figure 6: The non-heating parameters for $g_0 n_0 T = 2$. We consider all rational $\alpha = p/q$ with $q \leqslant 40$. Black dashed lines correspond to $I = 0$.

which leads to

$$\frac{p_1}{q_1} = \frac{1}{7}, \quad \frac{p_2}{q_2} = \frac{15}{106}, \quad \frac{p_3}{q_3} = \frac{16}{113}, \quad \frac{p_4}{q_4} = \frac{4687}{33102} \quad \dots \quad (50)$$

In Fig 5 (a), we plot the density of $\log(\lambda_{\mathbf{k}} T)$ as a function of two parameters $(g_0 n_0 T, k^2 T)$. In Fig 5 (b), we show a specific cut at $g_0 n_0 T = 2$ and demonstrate the expectation that the regime with zero heating rate $\lambda_{\mathbf{k}} = 0$ forms a Cantor set of zero measure.[9] We further zoom in a small regime in Fig 5 (c) and show that the self-similarity feature that was observed in Fibonacci driving is absent in the $\pi$-driving. On the contrary, what we see here in the $\pi$-driving is that the heating rate shows different patterns at different scales.

To further understand the formation of the structure at different scales, we show in Fig 5 (d) the heating rate distribution with $p_1/q_1 = 1/7$, namely the first principal convergent that approximates the irrational $\alpha = \pi - 3$. As excepted, the large-scale structure in Fig 5 (b) is captured while details are lost. In Fig 5 (e) and its zoom (f), we use the third principal convergent $p_3/q_3 = 16/113$, and find the patterns are well reproduced, even at the scale with magnitude of $\lambda_{\mathbf{k}} \sim 0.01$.

A simple understanding of the above phenomenons can be achieved via analogy with the band-theory of lattice Hamiltonian. Starting from some $p_{n-1}/q_{n-1}$, we have a band structure where the in-gap states correspond to the heating phase. When we increase $n$, the band folds, and new gap is opened. Intuitive, after we repeat this for infinite many times, the $\lambda_{\mathbf{k}} = 0$ regime (corresponds to the bands) breaks into a Cantor set. Now, from the definition (34), we know that when some $a_n$ is very large, $q_n$ is much larger than $q_{n-1}$. Consequently, $p_{n-1}/q_{n-1}$ becomes a particularly good approximation to $\alpha$ at a certain magnitude scale: the correction when taking $a_n$ into account comes from folding the Brillouin zone $a_n$ times, which is expected to be small when $a_n \gg 1$.

### 4.1.3 Hofstadter Butterfly

Knowing that the system is almost always in the heating phase for an irrational $\alpha$, we now give an overall picture for the phase diagram with varying $\alpha \in (0, 1)$. In Fig 6, we fixed $g_0 n_0 T = 2$ and draw all the non-heating region with $\alpha = p/q$ and $q \leqslant 40$. The result resembles the

---

[9]Generally, we expect if the integral of motion $I > 0$, under the trace map (39), almost all initial conditions lead to exponential growth, i.e. heating phase. In this case, it is hard to compute $I$ analytically. We numerically test that we have $I > 0$ for almost all parameters.

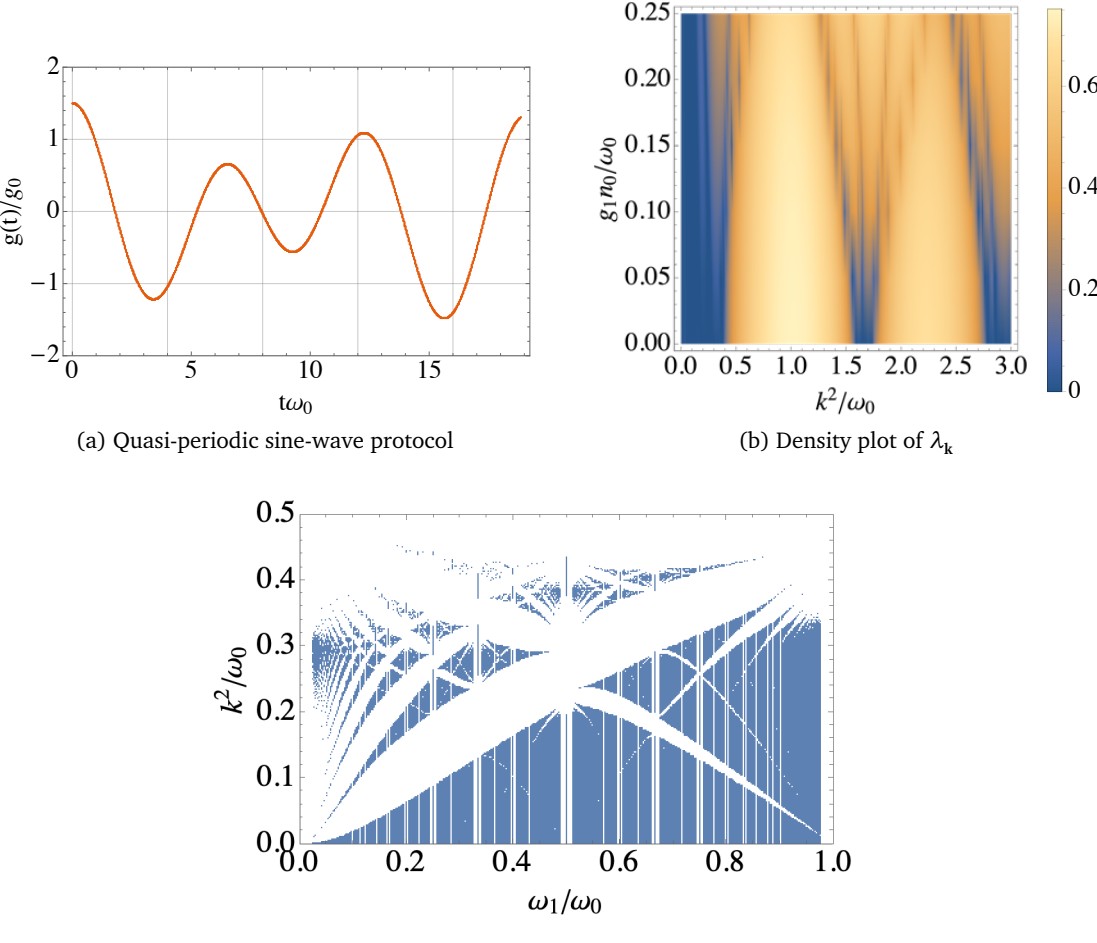

(a) Quasi-periodic sine-wave protocol

(b) Density plot of $\lambda_{\mathbf{k}}$

(c) Butterfly for sine-wave protocol

Figure 7: (a). The quai-periodic sine wave modulation of the scattering length. Here we set $g_1/g_0 = 0.2$ and $\omega_1/\omega_0 = \frac{\sqrt{5}-1}{2}$. (b). A density plot of $\lambda_{\mathbf{k}}$ with $g_0 n_0/\omega_0 = 0.5$ and $\omega_1/\omega_0 = \frac{\sqrt{5}-1}{2}$. (c) The non-heating parameters for $g_0 n_0/\omega_0 = 0.5$ and $g_1 n_0/\omega_0 = 0.1$. We consider all rational $\omega_1/\omega_0 = p/q$ with $q \leqslant 40$

Hofstadter butterfly [48]. Note that in our set-up, there is no reflection symmetry $\alpha \to 1 - \alpha$, due to the asymmetry between $U_0$ and $U_1$.

The existence of such a butterfly is as expected. Reminding the discussion in section 3.1, the square-wave protocol is an analogy of tight-binding models. In the current quasi-periodic setup, the corresponding tight-binding model takes the form of the Fibonacci model for the one-dimensional quasi-crystal [42,43]. Since both the Fibonacci model and the Aubry-André model [44] are quasi-crystals, and the Aubry-André model is directly related to the Harper model, where the original Hofstadter butterfly was discovered, by dimensional reduction. We expect there exists an analogy of the Hofstadter butterfly for the driven BEC case, as we observed in Fig 6.

## 4.2 Protocol 2: sine-wave

Now we turn to the second type of quasi-periodic driving. We consider $g(t) = 2g_0 \cos(\omega_0 t) + 2g_1 \cos(\omega_1 t)$. The driving is quasi-periodic when $\omega_1/\omega_0$ is an irra-

tional number. Again, the evolution is governed by (24), which we copy here:

$$\frac{d^2(\alpha_{\mathbf{k}}(t)-\beta_{\mathbf{k}}(t))}{dt^2} + \frac{k^2}{2}\left(\frac{k^2}{2} + 2g(t)n_0\right)(\alpha_{\mathbf{k}}(t)-\beta_{\mathbf{k}}(t)) = 0, \tag{51}$$

with the initial condition

$$(\alpha_{\mathbf{k}}-\beta_{\mathbf{k}})(0) = 1, \qquad \frac{d(\alpha_{\mathbf{k}}-\beta_{\mathbf{k}})}{dt}(0) = -i\left(\frac{k^2}{2} + 2g(t)n_0\right). \tag{52}$$

The model now corresponds to a 1D quantum mechanics in quasi-periodic potential [36, 37], also known as quasi-periodic Mathieu's equation in applied mathematics, e.g. see [38] for a review. When $g_0$ is much larger than $g_1$, one could take the tight-binding limit for small $k^2$, which gives the Aubry-André model.

Now we fix an irrational $\omega_1/\omega_0 = \frac{\sqrt{5}-1}{2}$ and $g_0 n_0/\omega_0 = 0.5$. We numerically solve (51) and fit the long time behavior to get the heating rate $\lambda_{\mathbf{k}}$. The result is shown in Figure 7 (b). For $g_1 = 0$ without quasi-random potential, we have non-heating phases around $k^2/\omega_0 = 0, 1.6, 3$, which corresponds to bands in a sine potential. When $g_1$ becomes finite, we expect a finite $k^2 g_1$ is needed for states to heat up. This is consistent with the existence of extended blue regions in Figure 7 (b) around $k^2 = 0$ and $g_1 = 0$.

Interestingly, we also see strip structures around the non-heating regions. To further explore structure, we focus on $g_1 n_0/\omega_0 = 0.1$ and change different $\omega_1/\omega_0$. We consider different rational numbers $\omega_1/\omega_0 = p/q$, and plot all the non-heating regions on the $\omega_1 - k^2$ plane. The result is shown in 7(c) which resembles some feature of the Hofstadter butterfly [48] for small but finite $k^2$, as expected from a naive tight-binding limit.

## 5 Summary

In this paper, we study two protocols of the periodically and quasi-periodically driven dynamics of Bose-Einstein condensates. We determine the phase diagram in terms of whether the system is heated or not, whether the heating phase has an exponentially growing number of excitations. For the quasi-periodically case with a square-wave modulation, phase with $\lambda_{\mathbf{k}} = 0$ forms a Cantor set of measure zero in the parameter space. We also exam a special non-heating point and find the number of excitations grows algebraically instead of exponentially. On the contrary, for the sine-wave quasi-periodic driving case, there is a finite measure regime where the system is in the non-heating phase. We also find analogs of the Hofstadter butterfly for both protocols. We expect that these experiments can be carried out with minor modifications using the experimental system in [15].

Finally, we would like to remark on the validation of the Bogoliubov approximation. By keeping the condensate wave function as a constant, we are assuming the number of particles in the initial condensate is much larger than the number of excitations. When the system is in the heating phase, this would ultimately break down. However, since this only happens in the late time, one expects the phase boundary receives little correction. This is similar to the CFT case [30], where the validation is checked by comparing the result to a lattice simulation. To estimate the time window for observing the exponential growth of the number of excitations, we consider the example of square-wave modulation for the periodically driving case with $T_0 = T_1$, $g_1 = 0$, and $g_0 n_0 T_1 \sim O(1)$. As we discussed in the section 3.1, for $k^2 \gg g_0 n_0$, excitations are hard to get excited. Consequently, we expect the total number of excitations proportional to $\sqrt{g_0 n_0}^3 \exp(\bar{\lambda} t)$, with some averaged heating rate $\bar{\lambda}$. We recognize the prefactor is the same as the quantum depletion in the thermal equilibrium. The time window to observe the exponential heating for some momentum $\mathbf{k}$ is then given by $1/\lambda_{\mathbf{k}} \lesssim t \lesssim \log(n_0^{-1/2} g_0^{-3/2})/\bar{\lambda}$, which is possible when the interaction $g_0$ is weak.

# Acknowledgment

We acknowledge Ruihua Fan, Ashvin Vishwanath, Xueda Wen and Zhigang Wu for discussions. P.Z. acknowledges support from the Walter Burke Institute for Theoretical Physics at Caltech. Y.G. is supported by the Gordon and Betty Moore Foundation EPiQS Initiative through Grant (GBMF-4306), and the US Department of Energy through Grant DE-SC0019030.

# A   The derivation of the trace map

In this appendix, We give the derivation of the trace map (39) for reader's convenience. The formula was obtained in [45] for special value $\alpha = \frac{\sqrt{5}-1}{2}$, and in [42] for general irrational numbers. The presentation here follows closely the one in [43], but with slightly different conventions.

We start with a lemma reducing powers of a rank 2 matrix with determinant 1.

**Lemma.** For a $2 \times 2$ matrix $\mathcal{U}_{\mathbf{k}}$ with determinant 1, denoting $x = \mathrm{Tr}(\mathcal{U}_{\mathbf{k}})/2$, we have

$$(\mathcal{U}_{\mathbf{k}})^n = S^{n-1}(x)\mathcal{U}_{\mathbf{k}} - S^{n-2}(x), \tag{53}$$

where

$$S^{n-1}(x) = \frac{\sin(n \arccos x)}{\sin(\arccos x)} \tag{54}$$

is the Chebyshev polynomial (of second kind). It also has an equivalent definition via recursion

$$S^n(x) = 2t S^{n-1}(x) - S^{n-2}(x), \quad S^0(x) = 1, \quad S^1(x) = 2x. \tag{55}$$

**Proof.** First, we consider $n = 2$ ($n = 1$ is trivially satisfied). We have $S^1(x) = x$ and $S^0(x) = 1$. Using Cayley–Hamilton theorem, we have

$$\mathcal{U}_{\mathbf{k}}^2 = 2x\mathcal{U}_{\mathbf{k}} - 1, \tag{56}$$

that is to say (53) is true for $n = 2$. Now we assume the lemma is true for all $n \leqslant a$, then

$$\begin{aligned}
(\mathcal{U}_{\mathbf{k}})^{a+1} &= S^{a-1}(x)(\mathcal{U}_{\mathbf{k}})^2 - S^{a-2}(x)\mathcal{U}_{\mathbf{k}} \\
&= S^{a-1}(x)(2x\mathcal{U}_{\mathbf{k}} - 1) - S^{a-2}(x)\mathcal{U}_{\mathbf{k}} \\
&= S^a(x)\mathcal{U}_{\mathbf{k}} - S^{a-1}(x).
\end{aligned} \tag{57}$$

Here we have used the recursion relation for the Chebyshev polynomial (55).

We now use the lemma to prove the recursion relation (39) (copied here)

$$\begin{aligned}
x_{\mathbf{k}}^n &= S^{a_n}(y_{\mathbf{k}}^{n-1})x_{\mathbf{k}}^{n-1} - S^{a_n-1}(y_{\mathbf{k}}^{n-1})z_{\mathbf{k}}^{n-1}, \\
y_{\mathbf{k}}^n &= S^{a_n-1}(y_{\mathbf{k}}^{n-1})x_{\mathbf{k}}^{n-1} - S^{a_n-2}(y_{\mathbf{k}}^{n-1})z_{\mathbf{k}}^{n-1}, \\
z_{\mathbf{k}}^n &= y_{\mathbf{k}}^{n-1}.
\end{aligned} \tag{58}$$

The third equation here is merely a definition, we just need to prove the first two equations. We have

$$\begin{aligned}
\mathcal{U}_{\mathbf{k}}(q_n T) &= \mathcal{U}_{\mathbf{k}}(q_{n-1}T)^{a_n}\mathcal{U}_{\mathbf{k}}(q_{n-2}T) \\
&= \left(S^{a_n-1}(y_{\mathbf{k}}^{n-1})\mathcal{U}_{\mathbf{k}}(q_{n-1}T) - S^{a_n-2}(y_{\mathbf{k}}^{n-1})\right)\mathcal{U}_{\mathbf{k}}(q_{n-2}T).
\end{aligned} \tag{59}$$

Taking the trace leads to $y_{\mathbf{k}}^n = S^{a_n-1}(y_{\mathbf{k}}^{n-1})x_{\mathbf{k}}^{n-1} - S^{a_n-2}(y_{\mathbf{k}}^{n-1})z_{\mathbf{k}}^{n-1}$. Similarly, we consider

$$\begin{aligned}
\mathcal{U}_{\mathbf{k}}(q_{n-1}T)\mathcal{U}_{\mathbf{k}}(q_n T) &= \mathcal{U}_{\mathbf{k}}(q_{n-1}T)^{a_n+1}\mathcal{U}_{\mathbf{k}}(q_{n-2}T) \\
&= \left(S^{a_n}(y_{\mathbf{k}}^{n-1})\mathcal{U}_{\mathbf{k}}(q_{n-1}T) - S^{a_n-1}(y_{\mathbf{k}}^{n-1})\right)\mathcal{U}_{\mathbf{k}}(q_{n-2}T),
\end{aligned} \tag{60}$$

whose trace leads to $x_{\mathbf{k}}^n = S^{a_n}(y_{\mathbf{k}}^{n-1})x_{\mathbf{k}}^{n-1} - S^{a_n-1}(y_{\mathbf{k}}^{n-1})z_{\mathbf{k}}^{n-1}$.

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
