# Peer review of "Periodically and Quasi-periodically Driven Dynamics of Bose-Einstein Condensates"

_SciPost Physics, doi:SciPost Phys. 9, 079 (2020)_

## Round 2 · Referee Report · Anonymous · 2020-9-17

Strengths

1. The Authors study BECs for a (quasi-)periodically modulated Bogoliubov scattering length.
2. They motivate the study by arguing that periodic drives can be used for many useful applications, but that the resulting quantum dynamics is difficult to solve.
3. There are some cases where symmetry allows for a large simplification, and they go on to consider such systems.
4. They draw a topological connection by generating a Hofstadter butterfly analogue for the two protocols under investigation.
5. Generally speaking, this appears to be novel, high quality work that adresses an important question.

Weaknesses

1. The most pressing issue for me is that a seemingly crucial connection is left unexplained.
It is not clear to me why when the system evolves as $n_k(t) \sim e^{\lambda_k t}$ we should consider it to be in a heating phase. That is, the connection between the physical system under study and the formalism has not been made clear.

2. There is a missing reference after 'non-trivial dynamics'

3. Some of the sentences are confusingly worded or gramatically incorrect.

Report

Overall, I think this work is of a high standard and I would be happy to recommend it for publication in SciPost Physics.

Requested changes

1. Physical justification for the criteria $n_k \sim e^{\lambda_k t}$ corresponding to the system being in the heating phase.

2. Fixing missig reference

3. Editorial assistance to fix grammar/wording

---

## Round 2 · Referee Report · Anonymous · 2020-9-28

Strengths

1) Heating of time-periodically driven interacting quantum systems is a timely subject that is relevant for experiment with atomic quantum gases.

2) The authors present (semi)analytical results for rather complex driving scenarios, such as quasiperiodic driving, which are non-trivial.

3) The Hofstadter-butterfly type pattern found when plotting the heating rates versus momentum and frequency ratio is a beautiful result, which is potentially observable in experiment.

Weaknesses

1) The main strategy for solving the dynamics of the Bogoliubov system, based on Eq. (7) and the observation that the problem can be formulated in terms of SU(1,1) operators, was developed already in previous work (which is properly cited).

2) The main results are statements about the asymptotic long-time dynamics computed within Bogoliuobov theory. However this theory equally breaks down in the long-time limit, when the condensate is depleted. However, the (existence of a) time window, where the asymptotic regime is reached while the Bogoliubov theory is still valid, is not discussed.

3) There are many grammatical errors.

Report

The authors describe heating in Bose-Einstein condensates that are periodically driven in time by a modulation of their interaction parameters. Such a modulation can be achieved experimentally in systems of ultracold atoms by exploiting a Feshbach resonance. Using the quadratic Bogoliubov approximation to the full Hamiltonian and assuming a fully condensed initial state, the long-time behaviour of the occupation of finite momentum modes are computed semi-analytically. In doing so four different driving scenarios are considered that can be devided into step-wise versus sinusoidal driving and time periodic versus time quasiperiodic driving.

While the main formalism is not new, the authors are able to derive analytical expressions for rather complex driving scenarios. In particular, for quasiperiodic driving they find a Hofstadter-butterfly type pattern, when plotting (scaled) heating rates versus momentum and a frequency ratio. This is a beautiful result that in principle could be investigated experimentally.

Before recommending publication, I would, however, like the authors to address the following point:

What is missing is a thorough discussion of the parameter regime, in which the theory is actually valid. Namely, on the one hand all calculations are done using Bogoliubov theory, which is valid only up to times, where the depletion is small compared to the total particle number. On the other hand, within this approximation statements about the asymptotic long-time limit are made. Thus, the authors should estimate both time scales, above which the asymptotic theory works and below which the Bogoliubov approximation breaks down, and then identify the parameter regimes, where the latter time is much shorter than the former.

Requested changes

1) There is a recent experiment in Hanns-Christoph Naegerl's group where the scattering length of ultracold atoms are modulated in time, which should be mentioned.

2) Eliminate grammatical errors.

3) Estimate the time scales for the breakdown of the Bogoliubov theory on the one hand and for reaching the asymptotic regime where the derived heating rates are valid on the other. Discuss the existence/duration of a time-window in which the derived results apply.

---

## Round 2 · Referee Report · Anonymous · 2020-10-20

Strengths

1) Good intuitive discussion of the background for their theoretical approach.
2) Makes quantitative theoretical contributions which lie at the intersection of two interesting topics of current experimental research: modulation of the scattering length and quasiperiodic driving.
3) Interesting and well-discussed exploration of complex features arising from quasiperiodic drives.

Weaknesses

1) The motivation for the phase-shifted periodic modulation drive protocol is not very clearly explained.
2) Some grammatical errors, misspellings, and stylistic oddities.
3) Too-brief discussion of some topics (especially butterfly-like graphs).

Report

The results are interesting and fairly well presented, and may be of broad interest as well as experimental relevance. In general I would recommend publication. Overall I am in broad agreement with the conclusions of the other reviewers.

The discussion of many of the complex features arising from quasiperodic driving shows good taste and opens a number of interesting directions for future research. A few things are touched on a little too briefly for my taste: the "Hofstadter butterfly" plots, for example, are shown but not discussed much at all. If they are going to be called that, a little discussion of the relation to the actual Hofstadter butterfly is in order.

Requested changes

1) A read-through for grammar and clarity of style.
2) A bit more discussion (just a few sentences) of the relationship between the graphs of non-heating parameters they show and the usual Hofstadter butterfly (eigenvalues of 2DEG in a magnetic field or just Harper model).
3) I agree with the other reviewer who requested a bit more discussion of the range of validity of their approach.

---

## Round 3 · Referee Report · Anonymous (Referee 1) · 2020-11-8

Report

Following the updating of this manuscript, I believe it should be published in scipost physics.

---

## Round 3 · Referee Report · Anonymous (Referee 3) · 2020-11-9

Report

The changes detailed by the authors sound appropriate; I support publication.

---

## Round 3 · Referee Report · Anonymous (Referee 2) · 2020-11-12

Report

The authors have addressed the points raised in my previous report in a satisfactory way. Therefore ,I recommend publication of the manuscript in its present form.

---

## Round 3 · Author Response

Dear editor,

Thank you and all referees for carefully reviewing our manuscript. We are happy to see referees find our work interesting and novel. Now we have revised our manuscript according to your suggestions. Please see the "List of changes" for details.

Hope you find our revised manuscript satisfactory and would like to suggest its publication.

Bests,
Pengfei Zhang and Yingfei Gu

---

## Round 3 · List of Changes

1. Add a physical justification for the criterion n_k\sim e^{\lambda_k t} corresponding to the system being in the heating phase.

Reply: We have now added “Consequently, the kinetic energy of atoms with the corresponding momentum, which is defined as k^2n_k/2, grows exponentially in time, implying the system is being heated. ” below equation (9).

  1. Estimate the time scales for the breakdown of the Bogoliubov theory on the one hand and for reaching the asymptotic regime where the derived heating rates are valid on the other. Discuss the existence/duration of a time-window in which the derived results apply.

Reply: We have now added estimation of the time window in the last paragraph of the summary section. The result clearly shows such a time window exists when the interaction is weak.

  1. More discussion of the relationship between the graphs of non-heating parameters they show and the usual Hofstadter butterfly (eigenvalues of 2DEG in a magnetic field or just Harper model).

Reply: To explain the relation, we have added: 1). The relation between our square-wave protocol and tight-binding models at the end of section 3.1; and 2). Explanations for the existence of the butterfly in sections 4.1.3 and 4.2.

  1. Please fix the grammatical issues.

Reply: We have read through our manuscript carefully and tried our best to fix grammar errors.

  1. Add reference after 'non-trivial dynamics'

Reply: We have fixed problems for the reference here.

  1. There is a recent experiment in Hanns-Christoph Naegerl's group where the scattering length of ultracold atoms are modulated in time, which should be mentioned.

Reply: We have added the reference in footnote 3.

---

## Editorial Decision

published